# Genetic Diversity of Montenegrin Local Sheep Breeds Based on Microsatellite Markers

**DOI:** 10.3390/ani12213029

**Published:** 2022-11-03

**Authors:** Milan Marković, Dušica Radonjić, Minja Zorc, Milena Đokić, Božidarka Marković

**Affiliations:** 1Biotechnical Faculty, University of Montenegro, Mihaila Lalića 15, 81000 Podgorica, Montenegro; 2Biotechnical Faculty, University of Ljubljana, Groblje 3, 1230 Domžale, Slovenia

**Keywords:** local breeds, microsatellites, genetic diversity, population structure

## Abstract

**Simple Summary:**

Sheep production is a very important sector in the rural economy and food security of people in the mountains area of Montenegro. The local and autochthonous sheep breeds are very specific and represent a very valuable genetic resource. Many of these sheep breeds are characterized by a permanent decreasing trend of the population size, as well as uncontrolled crossing with other breeds. Consequently, some of them are at risk of extinction and need to be included in the program of ‘‘in situ’’ conservation. The characterization of these breeds is mostly based on the investigation of the morphometric and production traits. This study focused on the genetic characterization of all known Montenegrin breeds and populations by using microsatellite markers. The obtained results should provide an important step for the establishment of an adequate strategy for the conservation and sustainable use of the sheep population. The preservation of breeds makes an important contribution to the preservation of Montenegrin traditional products and contributes to the sustainable development of overall sheep production.

**Abstract:**

The Montenegrin sheep population mostly consists of local breeds and their crossbreeds that are very valuable from their genome preservation point of view. The aim of this study was the investigation of the genetic diversity of seven Montenegrin sheep breeds (Jezeropivska—JP, Sora—SOR, Zetska zuja—ZZ, Bardoka—BAR, Sjenička—SJ, Ljaba—Lj, and Piperska zuja—PIP) using 18 microsatellite sets of markers. The genotyping was done for 291 samples from seven populations using the multiplex amplification of sequences with polymerase chain reaction (PCR). The parameters of genetic diversity were estimated using several software tools. In total, 243 alleles were found, with a range of 6 to 25 by locus. The mean observed heterozygosity (Ho), polymorphism information content (PIC), and Fis values (fixation index) per marker were 0.728, 0.781, and −0.007, respectively. The mean number of alleles per breed varied from 4.889 in ZZ to 10.056 in JP. The highest Ho was estimated for JP (0.763) and the lowest for ZZ (0.640). The genetic structure showed close relations between SOR and JP, and both of them with SJ, while ZZ, LJ, and PIP were more distanced. This study provides useful indicators for the development of further in-depth studies and the creation of appropriate conservation programs.

## 1. Introduction

The diversity of sheep breeds is the highest of all livestock species. There are about 1400 local and transboundary sheep breeds recorded in the Global Databank for Animal Genetic Resources [1]. The sheep population, especially that of local breeds, plays a very important role in the economic prosperity, food security, and cultural heritage of people in mountainous areas. In addition, the populations represent a potentially valuable genetic pool for many traits, including the adaption to extreme climate conditions, resistance to diseases, longevity, well adapted for walks over long distances, and grazing on poorly accessible terrain. [2,3]. In this line, the conservation and sustainable utilization of sheep genetic resources are of vital importance, especially in mountainous and less favorable areas for other livestock [4].

The current diversity of sheep breeds is influenced by various factors during the period of domestication, migration to new areas, the process of adaptation to different environmental conditions, and the application of selection and breeding programs [5,6]. Numerous studies on the genetic diversity of small ruminants have been performed. In earlier studies on sheep diversity and animal genetic resources, diversity was measured using morphological and biochemical markers such as blood groups and milk and blood protein polymorphism. However, in the last three decades, genetic diversity is usually analyzed by molecular markers [7,8]. The genetic characterization of local breeds is the first step in the prioritization of a breed for conservation and the development of appropriate conservation strategies [9].

Many molecular markers aid in the investigation of genetic diversity and structure as well as unravel the common genetic history of livestock populations. Numerous studies worldwide have been conducted in recent years on the genetic diversity and variability of local sheep breeds based on the analysis of microsatellite markers [10,11,12,13]. Microsatellite markers are highly polymorphic microsatellite markers, which are short sequence repeats of 1–6 base pairs. These types of markers, also known as Simple Sequence Repeats (SSRs), are the most widely used markers for the characterization of sheep populations to determine their purity or degree of crossbreeding with other breeds, as well as the genetic diversity within and among the population [14,15,16,17,18,19].

Sheep production is a very important branch of livestock production in Montenegro because it is mostly a mountainous area that is very suitable for sheep breeding. The total Montenegrin sheep population of 176,580 animals reared by about 5500 family farms and only a few companies [20] mostly consists of local (autochthonous) breeds and their crossbreds. All breeds belong to the group of coarse wool sheep with the common name “Pramenka” [21]. On the basis of morphometric and phenotype traits, seven different local breeds or populations were identified in Montenegro. Some of them are autochthonous breeds reared only in Montenegro, such as Jezeropivska, Zetska zuja, Sora, Piperska zuja, and Ljaba, while some of them are regionally transboundary breeds also reared in neighboring countries, such as the Sjenicka breed reared in Serbia and Bardoka (BAR) reared in Montenegro, Albania, Serbia, and Kosovo. All of these breeds are very well adapted to different rearing conditions and very valuable from their genome preservation point of view [22]. Given the decreasing trend of the total sheep population in Montenegro for a long time, the population of all of these breeds is decreasing [23]. Some of them are already very low in number and in danger of extinction (Zetska zuja, Ljaba, and Piperska zuja).

The present work aimed to investigate the genetic diversity and population structure of sheep breeds found in Montenegro. We estimated the genetic differentiation among and within representative populations through the analysis of the genetic variation based on microsatellite markers. We provide data on the spatial genetic structure of the Montenegrin sheep population in order to facilitate its sustainable development, utilization, and conservation.

## 2. Materials and Methods

### 2.1. Animals

Seven sheep populations reared in different geographic regions and climate conditions of Montenegro were used in the present study (Figure 1). Animals are selected according to their typical morphological breed characteristics described by references [21,22,23].

Jezeropivska sheep (JP) have coarse wool, are horned (male and female), have irregular black spots on their face, and are very well adapted for rearing in the very cold climate of the high mountain areas in Northwest Montenegro. The Sora breed (SOR) is a long-tailed breed with moderately coarse and white wool, black pointed face and legs, and horned (rams only), reared in the Northeast region. Bardoka (BAR) is a fully white-colored breed with very coarse wool, long fleece, and good milk production and is usually reared in the border region of Albania. Zetska zuja (ZZ) is at risk of extinction with a very small population of 100 to 200 breeding animals. It is a short-tailed sheep breed with a small body size, reddish-colored hair on the face and legs, and is very well adapted for rearing in the southern and central region, which is characterized by a very long dry and very hot summer season. Sjenicka sheep (SJ), with respect to body size, is the largest breed, mostly reared in the northern and central region of Montenegro, characterized by the black rings around the eyes, mouth, and half of the ears.

Ljaba (LJ) is a very small sheep, usually white colored with sporadically yellow spotted face and legs, reared in the Southeast region, which has Mediterranean climate conditions. Piperska zuja (PIP) is mostly reared in the central part of Montenegro; this breed has a medium body size, mostly horned (male and female), with yellow–red-colored head, ears, and legs. The most important body measures of the studied breeds are presented in Table 1.

### 2.2. Sampling

The biological material for the genotyping was collected from 291 animals of seven local sheep breeds: Jezeropivska (JP), 60 samples that were collected from four different flocks; Sora (SOR) and Sjenicka (SJ) breeds, 44 samples for both breeds collected from three flocks; Zetska Zuja (ZZ), 33 samples, Bardoka (BAR), 43 samples, and Ljaba (LJ), 44 samples, for all three breeds collected from two flocks; and 23 samples of Piperska zuja (PIP) that were collected from only one flock that was available as appropriate at that time. Blood samples were taken from animals that had morphological characteristics typical for that breed. Data and information provided by the owner of the farm were used for the selection of unrelated animals. Blood sampling was performed by taking 3 mL of blood out of the jugular vein into vacutainer blood collection tubes with EDTA anticoagulant. Samples were maintained at −20 °C until DNA extraction.

### 2.3. Laboratory Procedures and Microsatellite Quality Control

Genomic DNA was extracted from 200 μL of blood using following the method described by Ivankovic and Dovc [24]. The DNA concentration and purity were checked by a Nano-Vue Spectrophotometer taking the ratio of the optical density (OD) value at 260 and 280 nm, and then all samples were standardized to a DNA concentration of 50 ng/µL for further genotyping.

Eighteen microsatellite (MS) markers were used for genotyping purposes according to a recommendation of the Food and Agriculture Organization (FAO) http://www.fao.org/dad-is, accessed on 18 July 2022 [10] and the International Society for Animal Genetics (ISAG) https://www.isag.us/Docs/AppGenSheepGoat2017.pdf, accessed on 20 July 2022 [25] shown in Table 2.

The microsatellite genotyping was done by multiplex amplification of sequences using Polymerase Chain Reaction (PCR) using the primers recommended by ISAG in the thermocycler Dual 96-W Geneamp^®^ PCR System 9700 of Applied Biosystems in a total volume of 20 µL that contained 50 ng of DNA. The microsatellite amplification conditions were an initial denaturation cycle of 5 min at 94 °C followed by a denaturation step at 95 °C for 45 s in 35 cycles. Then, annealing was immediately performed at the recommended temperature (55°–65 °C) of each primer for 45 s, followed by the final extension step at 72 °C for 45s. After 35 repeated cycles, a final extension step at 72 °C for 10 min was conducted. Polymorphisms were analyzed through the electrophoresis of DNA fragments fluorescently labeled in the genetic analyzer 3500XL Capillary Genetic Analyzer of Applied Biosystems. The assignment of allelic polymorphisms was performed using specific genotyping GeneMapper Software version 4.1, Applied Biosystems, USA [26]. The analysis of the allele size and nomenclature followed the reference sample of the international comparison test ovine DNA ISAG 2013–2014. Estimates of allelic dropout, genotyping errors, false alleles, and null allele frequencies were assessed using Gimlet software version 1.3.3 [26] and with the FreeNA tool [27]. Deviations from Hardy–Weinberg equilibrium and linkage disequilibrium were tested using Genepop version 4.7.3 [28].

### 2.4. Data Analyses

The genetic diversity (the number of alleles, observed heterozygosity, unbiased expected heterozygosity, inbreeding coefficient) and deviations from the polymorphic information content (PIC) for each locus were estimated with the software Cervus v3.0.7 [29]. Uncorrected global F_ST_ was compared to F_ST_ values corrected using the excluding null allele method [30]. The effective population size (Ne) was estimated using NeEstimator v2.1 [31].

Principal component analysis (PCA) was performed using the ade4 library for R [32]. To assess the genetic structure of the seven sheep breeds, a Bayesian method was used. This analysis was performed using the model-based software STRUCTURE version 2.3.4 [33], which infers the number of genetic groups K present in a sample. The admixture model with 100,000 MCMC (Markov chain Monte Carlo) repetitions and 20,000 burn-in periods was used. Twenty independent runs were performed without prior information on groups assuming correlated allele frequency. K ranged from 2 to 14. The software Clumpak [34] was used for the visualization of the STRUCTURE results and the estimation of the optimal K value according to the Evanno method [35].

The UPGMA dendrogram was constructed on the basis of Nei’s standard genetic distance [36]. UPGMA was determined using MEGA 6 software [37].

## 3. Results

Eighteen microsatellite loci were amplified in seven Montenegrin sheep breeds, and all examined markers were found to be highly polymorphic in the whole population. The mean allelic dropout rate across loci was 0.160, the false allele rate was 0.331, and the mean error rate of the other five types of errors was 0.073.

### 3.1. Population Genetic Diversity of Microsatellite Locus

In total, 243 different alleles were found in the seven studied sheep breeds across 18 microsatellite loci. The lowest number of alleles (Na) was found at locus AE129a (six alleles), followed loci ILSTS011a and INRA049a, found in eight alleles, while the highest number of alleles was found at the locus CP49a (Na = 25), as presented in Table 3.

The null alleles (non-amplifying) showed frequency estimates ranging from 0.016 (INRA006a) to 0.056 (SPS115a). The frequency of null alleles for all alleles was particularly low. The mean null allele (No) across the 18 microsatellite loci was 3.1%.

The unbiased expected heterozygosis (uHe) as the most reliable parameter of genetic diversity in the population ranged from 0.598 (INRA172a) to 0.836 (CSRD247a), with a mean of 0.733 per loci, as the value of observed heterozygotes (Ho) varied from 0.573 to 0.863 with mean of 0.728

The polymorphic information content (PIC) was higher than 0.5 for all analyzed markers, ranging from 0.602 (INRA172a) to 0.938 (CP49a), so they could be considered highly informative. Global breed differentiation evaluated by F_ST_ ranged from 0.049 (INRA172a) to 0.093 (INRA023a). Seven markers significantly deviated from Hardy–Weinberg equilibrium (CSRD247a, ILSTS005a, INRA023a, INRA049a, INRA063a, SPS113a, and SPS115a).

According to the results presented in Table 4, the inbreeding coefficient F_IS_ varied from −0.065 to 0.046 per marker with a negative overall mean (−0.007), as presented in Table 4. Eight markers had negative F_IS_ estimates (CP49a, CSRD247a, FCB20a, INRA006a, INRA023a, INRA132a, MAF65a, McM042a, and SPS113a).

The total population F_IT_ and subpopulation F_ST_ values displayed positive values with mean values of 0.076 and 0.083. The average inbreeding coefficient of an individual related to the whole population (F_IT_) varied between 0.013 for SPS113a and 0.121 for AE129a, and the measurement of population differentiation (F_ST_) ranged from 0.059 (CSRD247a) to 0.106, obtained for FCB20a and INRA049a.

Gene diversity coefficient (Gst), total heterozygosity (Ht), and genetic diversity between populations were 0.072, 0.789, and 0.250, respectively.

### 3.2. Genetic Diversity between Sheep Breeds

The results of genetic diversity among breeds presented in Table 5 show that the lowest N_a_ per loci observed in the studied breeds was obtained for Piperska zuja (4.889) and the highest was obtained for Jezeropivska sheep (10.056). The effective number of alleles ranged from 3.195 (PIP) to 4.972 (JP). A very low number of alleles obtained for Piperska zuja might be explained by the small population and sample size compared with other breeds. The mean Shannon’s information index (I) ranged from 1.26 (PIP) to 1.77 (JP), with an average of 1.581. The mean value of the polymorphic information content (PIC) was higher than 0.7 for the Jezeropivska, Sora, Barkoka, and Sjenicka breeds, while the PIC values for the Zeta zuja, Ljaba, and Piperska breeds were 0.668, 0.669, and 0.650, respectively.

The mean of observed heterozygosity (H_O_) was 0.76, 0.75, 0.75, 0.69, 0.74, 0.72, and 0.69 for Jezeropivska, Sora, Zeta zuja, Bardoka, Ljaba, Sjenička, and Piperska zuja, respectively. The result shows a similar value of He for all breeds; however, for the LJ, SJ, and PIP breeds, expected heterozygosity was slightly lower than H_O._ The value of the inbreeding coefficient (F) observed per breed varied between −0.079 and 0.074, with a negative value for four breeds (ZZ, LJ, SJ and PIP).

ZZ and PIP had no deviation from HWE for any marker, whereas JP, SOR, and SJ deviated from HWE at only one marker, while the Bardoka (BAR) breed deviated for six markers (Appendix A).

### 3.3. Genetic Population Structure

Principal component analysis (PCA) revealed some differentiation between breeds ZZ, LJ, BAR, PIP (PC1, PC2), and SJ (PC3, PC4), but with some overlap among all seven breeds (Figure 2).

The PC1 and PC2 components were shown, of which the first axis explained 16.40% of the genetic variability of Piperska zuja and Ljaba separately from the five other breeds, and the second axis explained 13.67% of the variability of Zetska zuja separately from the other breeds. The individuals of most breeds were grouped together, which indicates admixtures among individuals. PC3 and PC4 accounted for 13.06% and 10.39% of the variability and showed a very admixed population, but one part of the population of Bardoka was separate.

In Figure 2, PC1 and PC2 separated the individuals of BAR, LJ, ZZ, and PIP into individual groups. The other three populations were positioned in the same place on the axis.

PC3 and PC4 showed that BAR had a wide distribution; one part of the population was positioned separately, and the other half was significantly admixed with the other breeds. This result was confirmed by structure analysis (Figure 3). Most of the individuals of the Sjenicka breed were positioned with those of the other breeds—Jezeropivska, Sor, and Bardoka (PC3 and PC4). The Jezeropivska breed has a wide distribution, so it was distributed compared to other sheep breeds in Montenegro.

The genetic population was determined based on the admixture level of each sheep individual. Each analyzed individual was represented by a single vertical line broken into colored segments (Figure 3). Bayesian clustering analysis recovered two genetic clusters, the optimal K based on Evanno was 2 (Figure 3). The value suggests that the studied sheep breeds were better defined by two genetic clusters. The first genetic cluster was made up of Jezeropisvka, Sora, Zetska zuja, and Sjenička sheep, with Bardoka, Ljaba, and Piperska zuja in the second cluster. Some individuals of Bardoka had high assignment probabilities with the first cluster.

The value k = 4 shows that the Bardoka and Piperska zuja populations with the broken color green and Zetska zuja (purple color) were separate. Seven separate clusters were recognized based on K = 7. The Jezeropivska breed had the highest proportion of the first gene pool (blue color). Based on the K value according to Evano, Figure 3 shows seven separate clusters, but some individual Sjenička, Ljaba, and Sora sheep had the same proportion of one gene pool. One part of the Bardoka individuals had a mixed genetic structure (colors from different clusters). The other part of the Bardoka flock had an admixture of all researched sheep breeds, which confirms the high level of historical mixing. Piperska zuja was geographically isolated in the northern part of Podgorica with very limited dispersal across the other regions of Montenegro.

The dendrogram of the studied sheep breeds (Figure 4) constructed on the basis of Nei’s standard genetic distance shows that JP and SOR were genetically close and belong to the same cluster; the nearest sub-cluster was SJ. Although separated, BAR and LJ were also quite close, and PIP and ZZ had the greatest genetic distances from the other breeds.

## 4. Discussion

The sheep population in Montenegro, mostly consisting of local breeds and populations, is under permanent pressure based on the decreasing trend of the population size, crossing with highly productive breeds or between local breeds, and selection without an appropriate breeding program or selection plan. All of this has affected some of the sheep breeds, especially those with low productivity, which are at risk of extinction and a reduction in their original genetic uniqueness [21,22,23].

In the present study, 18 microsatellite markers were used for the genetic characterization of the Montenegrin sheep population, genetic diversity (within breeds and between breeds), and the genetic structure of the investigated sheep populations. Most previous studies on the characterization of these sheep breeds is based on morphological traits [22,23], with very limited studies on the genetic characterization of only some of the breeds [12,38,39]. Consequently, it was not possible to compare most of the obtained results with other results for the same breeds.

Overall, the studied population of sheep showed a high level of diversity, as indicated by the number of alleles, the level of heterozygosity, polymorphic information content values (PIC), and other parameters of genetic diversity presented in Table 3 and Table 4. The determined average number of alleles per locus was 13.5, indicating a high diversity of alleles in the entire population. Since 15 of 18 used markers identified more than 10 alleles, and for two markers more than 20 alleles (CP49a and INRA063a), it can be regarded that a very informative set of markers was chosen. The mean N_A_ and the genetic variability of the Montenegrin sheep populations were similar to those reported for five sheep populations in Kazakhstan [17] and indigenous sheep breeds in Bulgarian and Turkey [40,41]. A higher N_A_ per locus was reported for 18 European sheep breeds [42], four Romanian sheep breeds, and 11 sheep breeds in India as well as Jordan sheep breeds [14,43,44]. Lower values of N_A_ were found by Salamon et al. [45] for 12 eastern Adriatic and western Dinaric native sheep breeds and 12 sheep breeds studied in Algeria [18].

Loci with more alleles are generally thought to produce more precise estimates of genetic distances than loci with few alleles, especially in studies on differentiation within livestock populations [14]. Three of the 18 investigated markers (CSRD247a, CP47a, and INRA133a) had Ho values higher than 0.8, while the overall mean of Ho for the total population was 0.728, ranging from 0.626 (AE129a) to 0.863 (CSRD247a). Fairly equal values of Ho and He, and for some markers even higher values of Ho than He, indicate a high diversity of the studied population of sheep but also the presence of interbreeding in the population. The population means of He and uHe were 0.723 and 0.733, respectively. These results are in accordance with results obtained for the group of autochthonous breeds from the Balkan Peninsula [12], Kazakh sheep breeds [17], some Croatian local breeds that belong to the group of coarse-wool Pramenka breeds [45], as well as Jordan and Arabian sheep breeds [43,46]. A higher diversity of the sheep population in terms of the average Ho and PIC was reported for 12 Algerian sheep breeds [18] and 14 Iran sheep breeds [47], and a lower diversity of the sheep population was obtained by many researchers [15,19,41].

The presence of null alleles is usually defined as non-amplifying alleles due to mutations at PCR primer sites, which have been associated with heterozygous deficits and causes overestimation of F_ST_ and genetic distance values. Only loci with a value of null alleles higher than 0.2 are considered potentially problematic for calculations of observed heterozygosity and inbreeding coefficients [18,45,46]. In our study, the frequency of null alleles was higher than 0.2 for 13 marker loci. However, a deep investigation of null alleles by Carlsson showed that null alleles have no major impact on the overestimation of F_ST_ and conclusions regarding the presence or absence of genetic differentiation [48].

The values of the polymorphism information content (PIC), generally suggest the polymorphic nature of the analyzed microsatellite loci [13]. All analyzed markers had PIC values greater than 0.5 or on average 0.781, which indicates a high diversity of the studied sheep population and the reliability of the used set of markers for the study of genetic diversity. The PIC value obtained in our study is higher than that reported by Girish et al. [11] and Salinas-Rios et al. [19] but lower than that reported for 12 Algerian sheep breeds [18].

The results of genetic diversity between the studied breeds in regard to the mean N_A,_ Ho, and He showed high genetic diversity in the most of studied breeds, except Piperska zuja (PIP) that had on average only 4.89 alleles and Ho and He lower than 0.7. Jezeropivska and Sora had the highest number of alleles per population (N_A_) and mean expected heterozygosity (He), indicating that they are the most genetically variable breeds. An investigation conducted by Cinkulov et al. [12] included JP and showed a higher value of N_A_ and He than our result, which indicates the decreasing diversity of these two breeds in the last 15 years. The higher average value of Ho than He identified for three breeds (ZZ, LJ, and PIP) may indicate a high level of variability among breeds [44]; however, this could indicate isolated braking effects, as reported previously [45]. In the case of our mentioned three breeds, it was possibly caused by the very small and isolated population. Slightly higher Ho than He in the SJ breed is not caused by a small and isolated population but most probably by crossing for the purpose of improving the production capacity. Similar results were reported for Creska and Privorska breeds that also have a relatively small population, as well as for Jordan and some Austrian breeds [8,44,45]. Many authors reported lower values of Ho and He per breed than those obtained in our study [7,9,15,42], as the other studies obtained a much higher level of heterozygotes and diversity [18,47]. A very high F value obtained for the Bardoka (BAR) breed (0.743), according to Oner et al. [49]. indicates an excess of homozygotes and can also cause deviations from HWE, which occurred in this breed.

Although the negative Fis value indicates that individuals in a population are less related than expected [5,6], the negative F value obtained in our study for three breeds (ZZ, LJ, and PIP) that are small and closed populations is not that relevant and indicates that null alleles probably obscure the real picture. The mean Fst value (0,083) explained that 8.3% of the genetic diversity was genetic variation between breeds and also indicated the existence of subpopulations. These results are similar to those obtained for Austrian and Mexico sheep breeds [8,15].

The genetic structure analyses provide the information needed for distinguishing breeds or populations if there is mixing in order to assign individuals or homogenous populations [19]. The results of structure analyses showed that the seven breeds clustered into two gene pools. Bardoka, Ljaba, and Piperska zuja sheep were assigned to a separate gene pool. The probable reason is that these breeds are small and geographically isolated and have relatively small populations. The second pool consisted of JP, SJ, and SOR, with a high level of overlapping.

The K value = 7 showed that SOR and SJ sheep share the same genome (same color), which may be the reason why SOR is crossed with SJ, so there are examples in selected samples. It also shows that BAR and LJ have a part of the genome that is recognized in SOR and SJ sheep. This may be due to crossbreeding with SJ sheep but also to the common gene shared by these breeds. This result is probably due to shared ancestry and also due to gene flow between the populations that are reared in close geographic areas. In general, these analyses confirm the wide distribution of the genome of the Jezeropivska and Sjenicka breeds, which is the actual situation.

## 5. Conclusions

This study of genetic diversity is the first one conducted on the whole Montenegrin sheep population. Genetic diversity within and among populations of seven sheep breeds was assessed using 18 microsatellite markers, which showed a high level of polymorphism. On the basis of the obtained results, all seven examined populations showed high genetic diversity through the effective number of alleles, relatively high mean number of alleles, heterozygosity, and high PIC values.

Structural analysis revealed the existence of two pools of breeds. The first consisted of JP, SOR, BAR, and SJ with a high degree of admixing, and the second was composed of three geographically isolated breeds with a small population (ZZ, LJ, PIP).

The results reported in this manuscript could provide useful indicators for the development of further in-depth studies by more carefully sampling animals and increasing sample sizes for some breeds to set appropriate conservation priorities, especially considering their vulnerability and potential economic and cultural importance.

## Figures and Tables

**Figure 1 animals-12-03029-f001:**
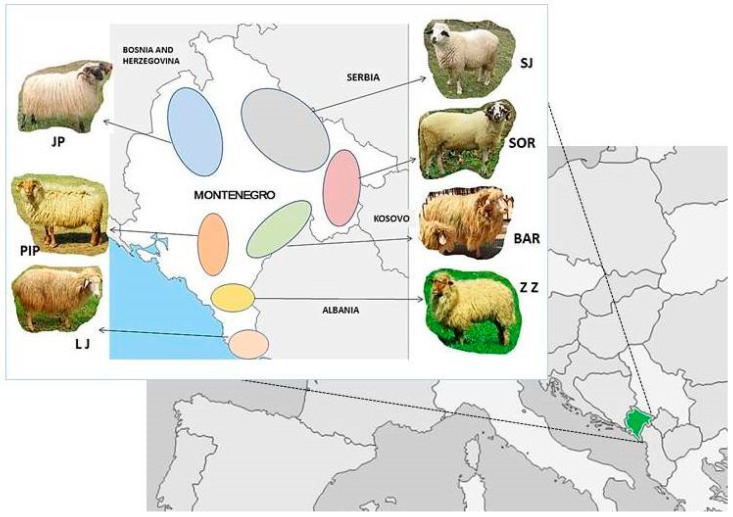
Locations of the breeding area of the Montenegrin sheep breeds.

**Figure 2 animals-12-03029-f002:**
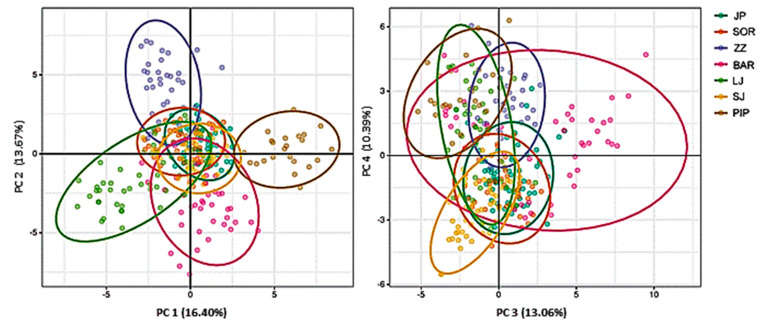
The PCA results of the studied sheep breeds (Abbreviations are listed in Table 5).

**Figure 3 animals-12-03029-f003:**
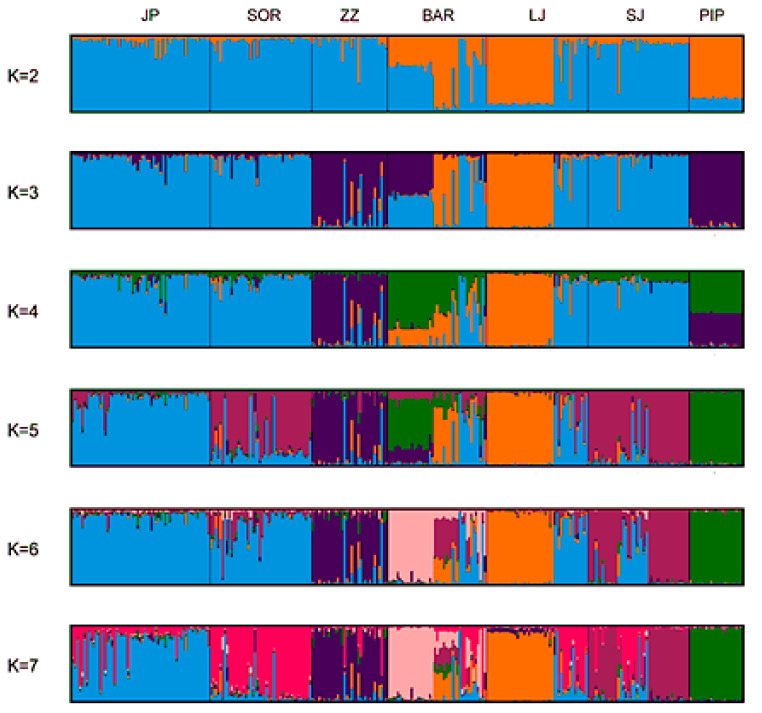
Clustering of sheep breeds by STRUCTURE.

**Figure 4 animals-12-03029-f004:**
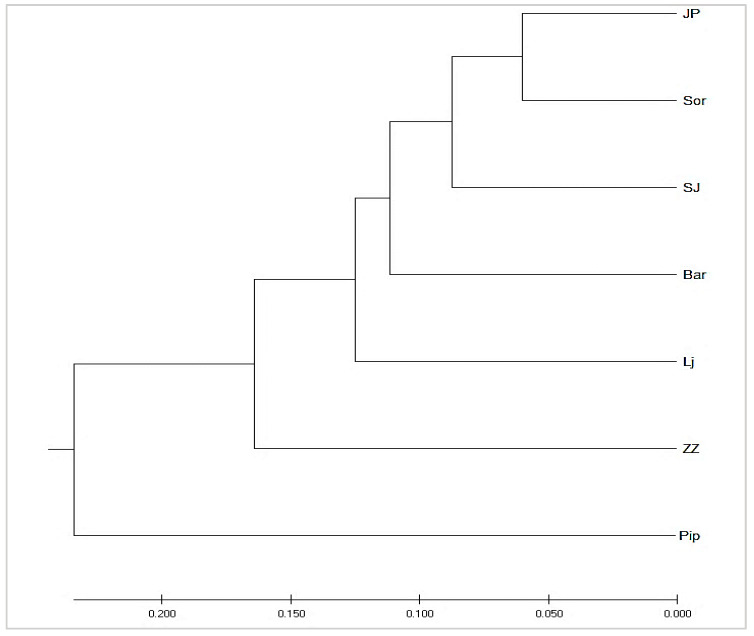
UPGMA dendrogram of seven populations of Montenegrin sheep constructed from Nei’s standard genetic distance (Abbreviations the same as those in Table 5).

**Table 1 animals-12-03029-t001:** The means of body measurements (cm) and body weight (kg) of sheep breeds. WH—wither height, BL—body length, CD—chest depth, CW—chest width, CC—chest circumference, CBC—cannon bone circumference, BW—body weight (Markovic et al., 2020).

Name of Breed	Acronym of Breed	WH	BL	CD	CW	CC	CBC	BW
Jezeropivska	JP	71.3	70.8	32.1	21.7	110.1	9.0	71.6
Sora	SOR	69.7	69.9	30.6	20.3	99.2	9.1	63.8
Zetaska zuja	ZZ	63.1	59.4	27.3	15.1	81.8	7.6	37.1
Bardoka	BAR	66.2	66.2	29.1	18.3	92.1	8.4	54.3
Ljaba	LJ	63.4	62.2	29.0	18.8	87.4	8.1	46.5
Sjenička	SJ	72.7	77.9	33.1	21.3	100.3	8.9	77.3
Piperska zuja	PIP	68.8	65.4	31.4	21.2	96.8	8.4	52.7

**Table 2 animals-12-03029-t002:** Information on the microsatellite markers used in the study (the microsatellite ID, genome position, annealing temperature, and allele length expressed in base pairs).

Marker	Chr.	Primer (5′–3′)	Annealing, °C	Size (bp)
OarAE 129	5	F: AATCCAGTGTGTGAAAGACTAATCCAG R: GTAGATCAAGATATAGAATATTTTTCAACACC	62	135–161
OarCP49	17	F: CAGACACGGCTTAGCAACTAAACGC R: GTGGGGATGAATATTCCTTCATAAGG	64	80–112
*CSRD247*	14	F: GGACTTGCCAGAACTCTGCAAT R:CACTGTGGTTTGTATTAGTCAGG	58	220–246
OarFCB20	2	F: GGAAAACCCCCATATATACCTATAC R: AATGTGTTTAAGATTCCATACATGTG	60	87–115
HSC	20	F: CTGCCAATGCAGAGACACAAGA R: GTCTGTCTCCTGTCTTGTCATC	63	263–297
ILST05	7	F: GGAAGCAATGAAATCTATAGCC R: TGTTCTGTGAGTTTGTAAGC	55	194–254
ILSTS011	9	F: GCTTGCTACATGGAAAGTGC R: CTAAAATGCAGAGCCCTACC	55	250–300
INRA006	1	F: AGGAATATCTGTATCAACCGCAGTC R: CTGAGCTGGGGTGGGAGCTATAAATA	64	110–132
INRA023	3	F: GAGTAGAGCTACAAGATAAACTTC R: TAACTACAGGGTGTTAGATGAACTC	58	194–216
INRA063	14	F: GACCACAAAGGGATTTGCACAAGC R: AAACCACAGAAATGCTTGGAAG	56	169–201
INRA049	1	F: TGTATTAGTTTGTGTTCTTTGGC R: TTGGCTTCCACAATCACACA	61	134–166
INRA132	20	F: AACATTTCAGCTGATGGTGGC R: TTCTGTTTTGAGTGGTAAGCT G	62	146–180
INRA172	22	F: CCAGGGCAGTAAAATGCATAACTG R: GGCCTTGCTAGCCTCTGCAAAC	65	126–172
MAF065	15	F: AAAGGCCAGAGTATGCAATTAGGAG R: CCACTCCTCCTGAGAATATAACATG	59	116–158
MAF214	16	F: AATGCAGGAGATCTGAGGCAGGGACG R: GGGTGATCTTAGGGAGGTTTTGGAGG	66	189–265
McM042	9	F: GTTCGTACTTCTGGGTACTGGTCTC R: GTCCATGGATTTGCAGAGTCAG	60	81–107
SPS113	10	F: CCTCCACACAGGCTTCTCTGACTT R: CCTAACTTGCTTGAGTTATTGCCC	60	126–152
SPS115	15	F: AAAGTGACACAACAGCTTCTCCAG R: AACGAGTGTCCTAGTTTGGCTGTG	62	246–260

**Table 3 animals-12-03029-t003:** Summary statistics of the seven sheep breeds with 291 animals genotyped in the present study. The number of alleles (Na), null allele frequency (No), observed heterozygosity (Ho), expected heterozygosity (He), unbiased expected heterozygosity (uHe), deviation from Hardy–Weinberg (HWD), *p*-value (ns = not significant, ** *p* < 0.01), inbreeding coefficient (F), and the polymorphic information content (PIC). Null allele frequency was estimated using the EM algorithm.

Locus	Na	No	Ho	He	uHe	HWD	F	PIC
AE129a	6	0.034	0.626	0.642	0.650	0.481 ^ns^	0.013	0.676
CP49a	25	0.026	0.816	0.794	0.804	0.942 ^ns^	−0.031	0.938
CSRD247a	17	0.017	0.863	0.825	0.836	0.003 **	−0.047	0.866
FCB20a	14	0.033	0.780	0.770	0.779	0.016 ^ns^	−0.014	0.918
HSCa	16	0.027	0.812	0.815	0.826	0.012 ^ns^	0.002	0.868
ILSTS005a	11	0.050	0.672	0.693	0.702	0.000 **	0.032	0.712
ILSTS011a	8	0.039	0.687	0.688	0.697	0.276 ^ns^	−0.012	0.724
INRA006a	11	0.016	0.723	0.685	0.694	0.196 ^ns^	−0.059	0.720
INRA023a	13	0.046	0.799	0.794	0.804	0.000 **	−0.010	0.876
INRA049a	8	0.040	0.673	0.677	0.686	0.000 **	0.012	0.704
INRA063a	21	0.039	0.744	0.758	0.768	0.000 **	0.027	0.794
INRA132a	14	0.024	0.833	0.815	0.826	0.025 ^ns^	−0.021	0.866
INRA172a	17	0.041	0.573	0.590	0.598	0.010 ^ns^	0.032	0.602
MAF214a	14	0.023	0.681	0.701	0.710	0.404 ^ns^	0.027	0.714
MAF65a	11	0.020	0.756	0.741	0.750	0.240 ^ns^	−0.026	0.770
McM042a	11	0.019	0.662	0.627	0.635	0.983 ^ns^	−0.062	0.843
SPS113a	12	0.000	0.680	0.638	0.646	0.002 **	−0.069	0.653
SPS115a	14	0.056	0.732	0.767	0.777	0.003 **	0.043	0.821
MEAN	13.5	0.031	0.728	0723	0.733	0.199	−0.009	0.781

**Table 4 animals-12-03029-t004:** Fixation index (Fis, Fit, Fst), genetic diversity among populations at each locus (G_ST_), total expected heterozygosity (Ht), and Jost’s estimate of differentiation (D).

Locus	Fis	Fit	Fst	Gst	Ht	D
AE129a	0.024	0.121	0.099	0.088	0.712	0.211
CP49a	−0.029	0.080	0.105	0.095	0.887	0.503
CSRD247a	−0.046	0.016	0.059	0.049	0.877	0.305
FCB20a	−0.013	0.095	0.106	0.096	0.861	0.439
HSCa	0.004	0.072	0.068	0.057	0.875	0.338
ILSTS005a	0.031	0.106	0.078	0.067	0.752	0.197
ILSTS011a	0.001	0.104	0.103	0.092	0.767	0.273
INRA006a	−0.055	0.027	0.078	0.068	0.743	0.193
INRA023a	−0.006	0.094	0.100	0.089	0.882	0.471
INRA049a	0.005	0.111	0.106	0.096	0.757	0.269
INRA063a	0.018	0.094	0.078	0.067	0.822	0.277
INRA132a	−0.022	0.051	0.071	0.060	0.877	0.355
INRA172a	0.029	0.091	0.063	0.052	0.630	0.096
MAF214a	0.028	0.093	0.066	0.055	0.751	0.167
MAF65a	−0.021	0.054	0.074	0.063	0.800	0.236
McM042a	−0.057	0.022	0.075	0.065	0.677	0.140
SPS113a	−0.065	0.013	0.074	0.064	0.689	0.145
SPS115a	0.046	0.128	0.085	0.074	0.839	0.326
MEAN	−0.007	0.076	0.083	0.072	0.789	0.250

**Table 5 animals-12-03029-t005:** Genetic diversity estimates grouped by breeds. Number of individuals sampled (N), mean number of alleles per population (N_a_), effective number of alleles per locus (N_E_), Shannon’s information index (I), observed heterozygosity (Ho), expected heterozygosity (He), unbiased expected heterozygosity (uHe), and inbreeding coefficient (F).

Breed	Acronym	N	N_a_	N_E_	I	Ho	He	uHe	F
Jezeropivska	JP	60	10.056	4.972	1.772	0.763	0.772	0.778	0.016
Sora	SOR	44	9.500	4.705	1.732	0.754	0.762	0.771	0.011
Zeta zuja	ZZ	33	7.722	3.708	1.515	0.747	0.704	0.715	−0.064
Bardoka	BAR	43	8.667	4.603	1.658	0.690	0.743	0.752	0.074
Ljaba	LJ	44	8.278	4.248	1.608	0.740	0.732	0.740	−0.015
Sjenička	SJ	44	7.889	3.703	1.522	0.717	0.711	0.719	−0.007
Piperska zuja	PIP	23	4.889	3.195	1.260	0.688	0.640	0.654	−0.079

## Data Availability

The data presented in this study are available in article and Appendix A.

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
