# Peer review of "Genetic Diversity of Montenegrin Local Sheep Breeds Based on Microsatellite Markers"

_animals, 2022, doi:10.3390/ani12213029_

Round 1

Reviewer 1 Report

The authors aimed to on the genetic characterization of all known Montenegrin breeds and populations by using microsatellite markers. The obtained results should provide an important step for the establishment of an adequate strategy for the conservation and sustainable use of the sheep population. The preservation of breeds makes an important contribution to the preservation of Montenegrin traditional products and contributes to the sustainable development of overall sheep production.

This manuscript can be accepted after major revision.

1. Line 67. Please give some information about the population size and number of animals per farm, and it may be better introduction giving some more such as doi: 10.34248/bsengineering.858274.

2. Line 81. Please give references.

3. Line 110. Figure 1 should be higher quality because the pictures cannot be recognized.

4. Line 118. “breeds are presented in Table S1” it is a small table not necessary as supplementary, it can be given as normal table.

5. Line 120. Is there any sample size estimation?

6. line 166. “[29Error! Reference source not found.].” ???.

7. Line 168. Could you give a sample code? It may be more attractive and increase the citation number.

8. Line 186. Instead of mean number of alleles per locus, giving “median: minimum and maximum numbers” are more informative because it is count data.

9. Line 254. Please give explanation rates and cumulative explanation a-rates of the components otherwise given information will be meaningless in statistical view.

10. Line 300. The dendrogram was interesting. There is no inner relations among breeds and the graph as stairs. It is confusing. Could you try another clustering methods such as Nearest Neighborhood or Ward?

11. Please add ethical consideration

Author Response

Dear Reviewer, 

 Dear reviewer,

Thank you very much for very useful comments and suggestions that are help us so much for  the improvement of the manuscript. We carefully passed all comments and revised the manuscript. 

Authors team.

Reviewer 2 Report

Although confined to a specific area (Montenegro), the work is of interest because it deals with the population study of several breeds of sheep. The methodology used is varied, although no novel methods are proposed that could make the work of broader interest. 

It had been of interest to include a cosmopolitan breed to compare it with the others, since as authors said, these locals breeds are crossed with highly productive sheep breeds.

Abstract

“243 alleles were found, … average number of 13.5 alleles per locus”. Information on the total number of alleles as well as the average number of alleles per locus is not informative (loci are very different), so my suggestion is to remove it from abstract. Range of number of alleles per locus (6 to 25) is informative. Also remove this information from results (3.1) and discussion sections.

Materials and Methods

How total number of animals within each population were chosen?. These numbers are a bit unbalanced, which could cause some bias when estimating similarity/relationships among them. Were those the only animals available or were sampled at random? Also, did the authors study the relationship between the individuals from the same breed? The latter could also conditioned the results of the degree of relationship among reeds. Authors said that unrelated animals were sampled (line 124) but the did not specified if that was for animals of the same breed (I suppose that). And what criteria was used to determine that animales were unrelated? Please add this information as it is important to understand the sampling process and the validity of the results.

Tables 2 is very informative, but numbers are average values for all breeds (please, indicate this in table legend). As authors know, when working with STRs average values are not usually representative. My suggestion is to include the same table but for each breed. These table could be as Supp. Material. If so, please discuss something about population parameter son each breed. Some STRs can be very informative in one breed but not in others.

Lines 210-213. Please, when starting to present results on a new parameter (Fst in this case), you have to indicate in which table this new parameter results are (Table 3 in this case).

Please, in Figure 2 authors should include percentage of variance explained by each principal component. When pesenting PC results, authors should included bree abbreviation together with breed name to make easy to understand PC figure.

Line 252. I do not see difference of breed SJ from others.

Tables

Table 2. The abbreviation used for observed heterozygosity (Ho) can be confused with homozygosity. I suggest authors to use other options, such as Het_obs and Het_exp for expected heterozygosity. Also in line 201 authors used uHE for unbiased expected heterozygosity (not mention before).

Figures

Figures resolution is poor and need to be improved.

Author Response

Dear Reviewer,

Thank you very much for  useful comments and suggestions that help us so much in  the improvement of the manuscript. We carefully passed all comments and revised the manuscript. 

Round 2

Reviewer 1 Report

Thak you for your valuable corrections.